# Greater Disability Is Associated with Worse Vestibular and Compensatory Oculomotor Functions in People Living with Multiple Sclerosis

**DOI:** 10.3390/brainsci12111519

**Published:** 2022-11-09

**Authors:** Colin R. Grove, Andrew Wagner, Victor B. Yang, Brian J. Loyd, Leland E. Dibble, Michael C. Schubert

**Affiliations:** 1Laboratory of Vestibular NeuroAdaptation, Department of Otolaryngology-Head and Neck Surgery, Johns Hopkins University, Baltimore, MD 21287, USA; 2Department of Otolaryngology-Head and Neck Surgery, The Ohio State University, Columbus, OH 43212, USA; 3School of Health and Rehabilitation Sciences, The Ohio State University, Columbus, OH 43212, USA; 4School of Medicine, Johns Hopkins University, Baltimore, MD 21287, USA; 5School of Physical Therapy and Rehabilitation Sciences, University of Montana, Missoula, MT 59812, USA; 6Department of Physical Therapy and Athletic Training, University of Utah, Salt Lake City, UT 84108, USA; 7Department of Physical Medicine and Rehabilitation, Johns Hopkins University, Baltimore, MD 21287, USA

**Keywords:** multiple sclerosis, video head impulse test, vestibulo–ocular reflex, compensatory saccade

## Abstract

Globally, there are nearly three million people living with multiple sclerosis (PLW-MS). Many PLW-MS experience vertigo and have signs of vestibular dysfunction, e.g., low vestibulo–ocular reflex (VOR) gains or the presence of compensatory saccades (CSs), on video head impulse testing (vHIT). We examined whether the vestibular function and compensatory oculomotor behaviors in PLW-MS differed based on the level of MS-related disability. The VOR gain, CS frequency and latency, and gaze position error (GPE) were calculated from the individual traces obtained during six-canal vHIT for 37 PLW-MS (mean age 53.4 ± 12.4 years-old, 28 females) with vertigo and/or an imbalance. The subjects were grouped by their Expanded Disability Status Scale (EDSS) scores: PLW-min-MS (EDSS = 1.0–2.5, *n* = 8), PLW-mild-MS (EDSS = 3.0–4.5, *n* = 23), and PLW-moderate-MS (EDSS = 5.0–6.0, *n* = 6). The between-group differences were assessed with Kruskal–Wallis tests. The VOR gains for most of the canals were higher for PLW-min-MS compared to PLW-mild- and mod-MS, respectively. CS occurred less often in PLW-min-MS versus PLW-mild- and mod-MS, respectively. No clear trend in CS latency was found. The GPE was often lower for PLW-min-MS compared to PLW-mild- and mod-MS, respectively. Thus, our data demonstrate that worse VOR and compensatory oculomotor functions are associated with a greater MS-related disability. PLW-MS may benefit from personalized vestibular physical therapy.

## 1. Introduction

Across the world, there are nearly three million people living with multiple sclerosis (PLW-MS) [1], a central nervous system disease that is characterized by the demyelination and inflammation of central nervous system neurons [2]. Nearly 75% of PLW-MS report experiencing dizziness, and the perception of disability due to dizziness can be classified as mild, moderate, or severe, in 60%, 31%, and 9% of these cases, respectively [3]. The odds of reporting moderate to severe dizziness increase with higher levels of self-reported disability in PLW-MS (severe vs. mild disability: odds ratio = 1.38) [3]. Additionally, both the impaired performance of the vestibulo–ocular reflex (VOR) and oculomotor dysfunction have been found in those with high levels of MS-related disability [4,5].

The majority of PLW-MS who report vertiginous sensations have compensatory saccades (CSs) on video head impulse testing (vHIT) [6,7,8], which suggests that the processing of semicircular canal (SCC) afference is disrupted. CSs are high acceleration eye rotations that are deployed by the central nervous system to rapidly reposition the eyes to reduce the gaze position error (GPE) that is induced by head rotations in the opposite direction [9]. As a pathologic sign of a reduced VOR gain, CSs can be detected during either the clinical head impulse test [10] or vHIT [11,12]. CSs have been observed during vHIT in healthy older adults [13], persons with peripheral vestibular hypofunction [14,15], and those with central nervous system diseases [16,17], including PLW-MS [7,10,11,12,18,19].

The lower the VOR gain, the greater the likelihood that CSs will be evoked and that CSs will have a larger amplitude and a shorter latency [13]. On average, the VOR gain is within normal limits (0.8 to 1.2) [20] in PLW-MS [7,19]. In response to lateral head impulses, PLW-MS have been shown to recruit CSs less often and with a slightly longer latency than persons who are status-post unilateral vestibular deafferentation [19] but more often with a reduced latency relative to healthy controls [21,22]. PLW-MS who have documented infratentorial lesions [23,24] and with suspected occult [18] internuclear ophthalmoplegia (INO) are reported to have asymmetrical abnormal VOR gains.

To date, only the VOR gain and CS frequency from six-canal vHIT have been reported in investigations of the vestibular function and compensatory oculomotor behavior in PLW-MS [7,23,24]. Additionally, few studies have investigated the association between the tests of the SCC function and MS-related disability [25]. Analyzing the vestibular function and compensatory oculomotor behavior in PLW-MS who report symptoms of dizziness may provide an insight into the association between vestibular dysfunction and disability in MS. Thus, the dual purposes of this study were (1) to characterize the visual–vestibular function in PLW-MS using six-canal vHIT results and (2) to compare the vestibular function and compensatory oculomotor behavior across groups of PLW-MS with differing levels of disability. We hypothesized that a greater MS-related disability would be associated with a reduced VOR gain, an increased CS frequency, a longer CS latency, and an increased GPE.

## 2. Materials and Methods

### 2.1. Participants

We analyzed the baseline data for 37 adults that were previously collected for an intervention study [26]. Commonly, the effects of MS on the neurological function are documented using Kurtzke’s Expanded Disability Status Scale (EDSS) [27]. Aside from documenting the presence of nystagmus, tests of the vestibular function are not included in the EDSS. Kurtzke described the levels of disability due to MS on the EDSS as 1.0 to 2.5 = minimal, 3.0 to 4.5 = mild, 5.0 to 6.5 = moderate, and 7.0 to 9.5 = severe [27]. This sample included three groups of PLW-MS: 8 people living with minimal MS (PLW-min-MS: EDSS = 2.0 to 3.5), 23 people living with mild MS (PLW-mild-MS: EDSS = 3.0 to 4.5), and 6 people living with moderate MS (PLW-mod-MS: EDSS = 5.0 to 6.0).

### 2.2. Inclusion and Exclusion Criteria

The diagnosis of MS was confirmed by a neurologist in all cases. Additional inclusion criteria were evidence of an impaired balance (a history of ≥2 falls in the past year, an Activities Balance Confidence Scale [28] average score <80%, or a total score on the Dynamic Gait Index [29] <19/24) and/or self-reported dizziness (Dizziness Handicap Inventory [30] >0/100).

The existence of co-morbid peripheral vestibular and central dysfunction was ruled out by recording a patient history and completing a bedside clinical exam with video-oculography. The examination included a spontaneous nystagmus, post-head-shaking nystagmus, the oculomotor function (i.e., saccades and smooth pursuit), a VOR suppression), a clinical yaw head impulse test, and tests for benign paroxysmal positional vertigo [31].

All the testing was conducted while each participant’s disease was stable, i.e., during a period in which there was no clinical evidence of MS exacerbation. PLW-MS who had a concurrent peripheral vestibular dysfunction, a central nervous system disorder in addition to MS, or EDSS scores > 6.0 were not eligible. The EDSS was administered by a trained assessor within one week of each subject’s enrollment into the study. No participants had clinical evidence of INO.

### 2.3. Video Head Impulse Test

vHIT was administered using the ICS Impulse system (Otometrics, Natus Medical Incorporated; Taastrup, Denmark). Data for all the canals/planes (left/right-lateral, right-anterior/left-posterior, and left-anterior/right-posterior) was collected. Participants viewed a target positioned on the wall at 1 m while the assessors delivered low amplitude (10–25°), high acceleration (4000°/s^2^) canal plane head impulses. The head movement data were collected with goggle-mounted accelerometers and the eye velocity was video-recorded at 250 frames/s. A custom-written MATLAB code (Natick, MA, USA) was used to process the raw vHIT data traces for each head impulse [18,19].

### 2.4. Identification of Compensatory Saccades

An acceleration threshold of 4000°/s^2^ was used to help identify each CS [13,32]; however, the onset and offset of each CS was defined using an iterative process. The onset of each CS was approximated by finding the peak eye acceleration (≥4000°/s^2^) and then the point where the acceleration initially rose above 2000°/s^2^ [33]. Next, the termination of each CS was identified by working forward to find the time after the deceleration curve was above −4000°/s^2^ and the eye stopped moving in the same direction [32]. In the next step, the deceleration peak was reset more conservatively at −2000°/s^2^ to help delineate a covert CS from the slow phase eye velocity (SPEV). The approximate offset point for a CS was set as the time when the deceleration peak was >−2000°/s^2^ and the eye acceleration ceased to monotonically decrease.

After defining the approximate boundaries for CSs, the onset and offset points of each CS could be refined. The eye velocity data were passed through a zero-phase loss, a 20 Hz low pass filter (MATLAB; filtfilt), to identify the peak velocity between the approximate CS onset and offset points. The filtered eye velocity data were then used to determine the first point where the CS velocity rose above 10°/s [34]. Next, the onset point for each CS was set as the intersection between two linear fits (MATLAB; polyfit) of the peak eye velocity, the first point when the eye velocity rose above 10°/s and a point 44 ms prior [34]. At the same time, the offset point for each CS was set as the intersection between two linear fits between the peak eye velocity and the first point after the peak eye velocity fell below 10°/s and a point of 44 ms later. Additionally, a minimum CS duration was set at 6 ms.

Each processed trace with the CSs marked was reviewed by two trained raters (CRG, VBY) who assessed the data for errors or artifacts, and then finalized the position of the onset and offset points for each CS. If the CS endpoints fell outside the anticipated parameters based upon a visual inspection of the eye velocity, acceleration, and/or position traces, then the onset and/or offset point(s) were approximated using the available data [13,18,19,32,35].

### 2.5. Variables of Interest

The VOR gain was calculated by dividing the area under the eye velocity curve (with covert CS removed) by the area under the head velocity curve [9,13,20,22,35,36]. Although normal values range from 0.80 to 1.20 on vHIT, a VOR gain of 1.0 reflects an optimal compensatory SPEV response relative to a given head velocity [20]. The overall CS frequency was calculated as the total number of all CSs per head impulse. We also calculated the overall CS latency of all CSs that occurred during each head impulse. The effectiveness of the CSs in overcoming the GPE was assessed by calculating the difference (in degrees) between the position of the eye at rest prior to the head impulse and the position of the eye immediately following the head impulse, at the point that the head velocity reached 0°/s. GPE is a more comprehensive measure of the functioning of the gaze stability system in the position domain as it reflects the combined influence of both the SPEV and any covert CSs [19]. To resolve a GPE of 4°, the nervous system must generate either a single CS with an amplitude of −4° or multiple CSs with the same combined amplitude that travel in the direction of the earth-fixed visual target.

### 2.6. Statistical Analysis

We conducted all analyses in R for Statistical Computing (v. 4.1.2) [37]. The a priori alpha level for each analysis was 0.05. Most metrics were not normally distributed (Shapiro–Wilkes tests) and their variances were not equal (Fligner–Killeen tests); therefore, the overall between-group differences were evaluated using Kruskal–Wallis tests. Additionally, we performed pairwise Wilcoxon Rank Sum tests to examine the differences between the specific groups, and we used Holm’s method to adjust the *p* values for pairwise comparisons. The participant characteristics of their age, sex, age at the onset of their MS, and the duration of their MS are presented in Table 1 as the mean (standard deviation) number (percent) and median (interquartile range [IQR] as appropriate. The overall between-group differences for the median (IQR) for the VOR gain, CS frequency, CS latency, and GPE for each SCC are presented in Table 2.

## 3. Results

### 3.1. Participant Characteristics

The data were analyzed from 37 PLW-MS. The mean age was similar for PLW-min-MS [47.3 (12.2) years], PLW-mild-MS [55.5 (11.9) years], and PLW-mod-MS [54.2 (13.8) years]. Each group included a similar proportion of female subjects [6 (75.0%) PLW-min-MS, 18 (78.3%) PLW-mild-MS, and 4 (66.7%) PLW-mod-MS]. The median age at the onset of MS was similar for PLW-min-MS [41.0 (33.8–47.2) years], PLW-mild-MS [32.0 (28.0–42.5) years], and PLW-mod-MS [38.5 (26.5–46.0) years]. The median disease duration was shorter for PLW-min-MS [2.0 (1.0–9.2) years] compared to PLW-mild-MS [21.0 (11.0–28.5) years, *p* = 0.026] and PLW-mod-MS [18.5 (13.5–19.8) years]. Based on their EDSS Functional Systems Scores, all but one participant had evidence of a brainstem dysfunction and all participants had evidence of a cerebellar dysfunction. We focus our presentation of the vHIT results on our pairwise analyses of the CS latency and GPE.

### 3.2. Right–Left Lateral Canal Findings

#### 3.2.1. Right Lateral Canal

Our analyses of the data from right-lateral canal impulses revealed between-group differences for the VOR gain (*p* < 0.001), CS frequency (*p* = 0.002), and GPE (*p* < 0.001), but not for the CS latency (*p* = 0.152). For the right-lateral impulses, the latency of CSs was similar for PLW-min-MS [230.00 ms (185.00–283.25 ms)], PLW-mild-MS [220.00 ms (168.00–296.00 ms)], and PLW-mod-MS [200.00 ms (160.00–257.33 ms)], *p* = 0.152. The GPE following the right-lateral impulses was larger for PLW-mild-MS [1.16° (0.10°–2.72°)] compared to PLW-min-MS [0.18° (−0.20°–1.01°), *p* < 0.001] and PLW-mod-MS [0.26° (−0.95°–1.57°), *p* < 0.001], but the GPE was similar for PLW-min- and PLW-mod-MS, *p* = 0.876.

#### 3.2.2. Left Lateral Canal

Our analyses of the data from the left-lateral canal impulses demonstrated between-group differences for the VOR gain (*p* < 0.001), CS frequency (*p* = 0.013), and GPE (*p* < 0.001), but not for the CS latency (*p* = 0.085). The latency of CSs in response to the left-lateral impulses was similar for PLW-min-MS [244.00 ms (210.33–308.00 ms)], PLW-mild-MS [256.00 ms (213.00–312.33 ms)], and PLW-mod-MS [276.00 ms (223.50–324.00 ms), *p* = 0.085. The GPE following left-lateral impulses was larger for PLW-mild-MS [2.09° (0.85°–3.60°)] compared to PLW-min-MS [1.30° (0.52°–1.97°), *p* < 0.001] and PLW-mod-MS [0.84° (−0.12°–1.81°), *p* < 0.001], yet the GPE was similar for PLW-min- and PLW-mod-MS, *p* = 0.088.

### 3.3. Right Anterior–Left Posterior Canal Findings

#### 3.3.1. Right Anterior Canal

Our analyses of the data from the right-anterior canal impulses showed between-group differences for the VOR gain (*p* < 0.001), CS frequency (*p* < 0.001), CS latency (*p* = 0.001), and the GPE (*p* < 0.001). The CS latency was longer with the right-anterior impulses for PLW-mild-MS [222.00 ms (140.00–280.00 ms)] compared to PLW-min-MS [140.00 ms (90.00–206.00 ms), *p* = 0.008] and PLW-mod-MS [174.67 ms (100.00–213.50 ms), *p* = 0.028], yet the CS latency was similar for PLW-min- and PLW-mod-MS, *p* = 0.573. The GPE was larger after the right-anterior impulses for PLW-mod-MS [3.50° (2.33°–4.31°)] compared to PLW-min-MS [1.43° (0.85°–2.69°), *p* = 0.003] and PLW-mild-MS 1.72° (−0.19°–3.44°), *p* < 0.001]; the GPE was similar for PLW-min- and PLW-mild-MS, *p* = 0.779.

#### 3.3.2. Left Posterior Canal

Our analyses of the data from the left-posterior canal impulses demonstrated between-group differences for the VOR gain (*p* < 0.001), CS latency (*p* = 0.001), and GPE (*p* < 0.001), but not for the CS frequency (*p* = 0.816). The latency of CSs evoked by the left-posterior impulses was longer for PLW-mild-MS [211.00 ms (172.00–275.50 ms)] compared to PLW-min-MS [166.00 ms (103.00–210.00 ms), *p* = 0.004] and PLW-mod-MS [184.33 ms (92.00–255.50 ms), *p* = 0.029], yet the CS latency was similar for PLW-min- and PLW-mod-MS, *p* = 0.431. The GPE following the left-posterior impulses was larger for PLW-mild-MS [4.32° (2.02°–6.75°), *p* < 0.001] and PLW-mod-MS [5.02° (2.77°–6.31°), *p* < 0.001] compared to PLW-min-MS [0.79° (−0.36°–1.77°)], but the GPE was similar for PLW-mild- and PLW-mod-MS, *p* = 0.534.

### 3.4. Left Anterior–Right Posterior Canal Findings

#### 3.4.1. Left Anterior Canal

Our analyses of the data from the left-anterior canal impulses revealed between-group differences for the VOR gain (*p* < 0.001), CS frequency (*p* = 0.002), CS latency (*p* = 0.001), and the GPE (*p* < 0.001). The latency of CSs following the left-anterior impulses was longer for PLW-mild-MS [228.00 ms (152.00–316.00 ms)] compared to PLW-mod-MS [159.67 ms (100.00–214.00 ms), *p* = 0.002]; however, the CS latency was similar between PLW-mod- and PLW-min-MS [197.00 ms (115.50–294.00 ms), *p* = 0.318] and between PLW-min- and PLW-mod-MS, *p* = 0.339. The GPE following the left-anterior impulses was larger for PLW-mild-MS [3.00° (1.19°–4.78°), *p* < 0.001] and PLW-mod-MS [3.18° (1.92°–5.37°), *p* < 0.001] compared to PLW-min-MS [1.65° (0.48°–2.30°)], yet the GPE was similar for PLW-mild- and PLW-mod-MS, *p* = 0.371.

#### 3.4.2. Right Posterior Canal

Our analyses of the data from the right-posterior canal impulses showed between-group differences for the VOR gain (*p* < 0.001), CS frequency (*p* = 0.03), and GPE (*p* < 0.001), but not for the CS latency (*p* = 0.086). The latency of CSs evoked by the right-posterior impulses was similar for PLW-min-MS [220.00 ms (130.00–271.00 ms)], PLW-mild-MS [209.33 ms (172.00–308.00 ms)], and PLW-mod-MS [200.00 ms (128.00–262.00 ms)], *p* = 0.86. The GPE following the right-posterior impulses was greater for PLW-mild-MS [4.99° (2.50°–8.53°)] compared to PLW-min-MS [2.84° (1.83°–3.61°), *p* < 0.001] and PLW-mod-MS [3.82° (2.57°–5.18°), *p* = 0.006], and the GPE was greater for PLW-mod- versus PLW-min-MS, *p* = 0.006.

### 3.5. Atypical Compensatory Oculomotor Behavior

We observed several instances of inappropriate responses to vHIT in PLW-mod-MS. Despite having abnormal VOR gains for a particular SCC, some participants failed to generate CSs in response to the head impulses (Figure 1). Two moderately disabled participants experienced an exacerbation of down beating nystagmus following vertical canal impulses (Figure 2). Another individual with a moderate disability generated a very large amplitude yet slow velocity compensatory eye movements in response to the vertical canal impulses (Figure 3). These oculomotor behaviors were not seen in less disabled participants.

## 4. Discussion

We sought to further characterize the visual–vestibular function in PLW-MS and to determine whether the vestibular function and compensatory oculomotor behavior are associated with MS-related disability. Our results demonstrate that a greater MS-related disability was associated with a reduced VOR gain and an increased GPE, particularly as it relates to head impulses in the planes of the vertical SCCs. However, the frequency with which CSs were generated to minimize the GPE and CS latency were variable across the levels of MS-related disability as measured by the EDSS. Thus, categorizing the severity of MS based on the EDSS score was only partially sufficient for providing a framework for describing the association between MS-related disability and the gaze stabilization mechanisms.

### 4.1. Association between Measures of Gaze Stability and Disability

Our findings of relatively well-preserved VOR gain and minimal GPE in response to horizontal (lateral) SCC impulses suggests that there can be a considerable sparing of the VOR canal function in PLW-MS, even in those with greater levels of disease-related disability. In a previous study in which we analyzed the lateral SCC vHIT results in PLW-MS and adults who were status-post vestibular deafferentation, we found that the VOR gain, although less than unity, was, on average, relatively preserved in PLW-MS [19]. These results are also in agreement with others who found normal values (0.80 to 1.20) for horizontal (lateral) SCC VOR gains in PLW-MS [7]. Our results related to the GPE are consistent with our previous report that PLW-MS were able to effectively minimize the GPE following horizontal impulses through a combination of preserved SPEV and the generation of appropriately compensatory saccades [19]. Given that the GPE following right- and left-lateral impulses was <1.0° for most participants, regardless of the level of MS-related disability, one might not expect that PLW-MS would experience significant oscillopsia related to rapid horizontal head movements; however, PLW-MS have been shown to perform >2.5 times worse than healthy adults on computerized dynamic visual acuity testing [21]. This difference may be due to the fact that we analyzed gaze stability after subdividing the study population by the level of MS-related disability rather than across the entire population [22]. Alternatively, since the head movement acceleration and frequency are much lower for the dynamic visual acuity testing compared to vHIT, this apparent discrepancy may suggest that PLW-MS experience a selective decline in lower frequency VOR responses.

Our findings which show that the VOR gains are markedly reduced and the GPE is large in response to the vertical head impulses also suggest that, in those with a greater MS-related disability, the disease preferentially affects the vertical canal pathways. Specifically, the posterior SCC pathways are more affected than the anterior SCC pathways, and both sets of vertical SCC pathways are more affected than the lateral SCC pathways. The GPE was large following the anterior and posterior canal impulses, particularly for PLW-mod-MS who had median GPEs of 3.18° to 5.02° following the vertical canal plane impulses. Thus, as MS-related disability advances, oscillopsia may become a significant safety issue during activities that involve high-frequency vertical head movements [38,39], e.g., walking, running, and driving.

The graduated effects of MS on the lateral, anterior, and posterior canal pathways that we describe differ from the prior findings in a younger cohort (mean age 19.06 ± 1.66 years old) with childhood-onset MS [6]. The median VOR gains for the lateral (0.96), anterior (1.00), and posterior (0.91) canals in those with childhood-onset MS are much higher than the VOR gains that we report; however, there was a significant difference in the median VOR gain for those with childhood-onset MS who did (0.91) and did not (1.01) report symptoms of dizziness [6]. These apparent discrepancies with our results may be explained by the fact that the participants’ mean age at the onset of their MS was higher, the median disease duration at the time of testing was longer, and the lesion burden may have been greater in the participants included in our cohort compared to the cohort of those with childhood-onset MS.

### 4.2. Effects of Internuclear Ophthalmoplegia on Video Head Impulse Test Results

Infratentorial involvement is common in PLW-MS as over 70% of affected persons have brainstem lesions on magnetic resonance imaging and nearly 90% have abnormal brainstem reflexes [40]. Signs of cerebellar involvement are also present in the majority of PLW-MS [41]. Lesions in the brainstem and cerebellum are associated with oculomotor dysfunction in 80% of PLW-MS [42]. Of those PLW-MS who report audio-vestibular symptoms, 37.5% have been found to have at least one active or enlarged (compared to prior imaging) brain lesion [43]. PLW-MS who report vestibular symptoms are more likely to have infratentorial lesions compared to supratentorial lesions [43].

Our findings in PLW-MS, nearly all of whom had clinical evidence of brainstem involvement, are in agreement with the extend prior studies that reveal a similar pattern of graduated involvement of the lateral, anterior, and posterior SCC pathways in adults who have evidence of INO associated with medial longitudinal fasciculus lesions on magnetic resonance imaging [23,24]. In a cohort of adults with brainstem infarcts or demyelination that was associated with INO, the results, which were obtained from a vHIT system not unlike the system we used, revealed that a VOR gain reduction was more pronounced for the contralesional posterior SCC than the ipsilateral lateral and anterior SCCs in unilateral INO [24]. It is of note that PLW-MS who have either unilateral or bilateral INO have been shown to have asymmetrical ipsilesional VOR deficits and that the VOR gain is lower for the adducting compared to the abducting eye when the eye movement is measured binocularly using search coils [23]. Additionally, Aw and colleagues showed that whenever there was a detectable INO, the contralateral posterior SCC VOR gain was markedly reduced to less than 50% of the normal level; however, the anterior SCC VOR gain was only moderately impaired [23]. The finding of a similar pattern of VOR gain results in our cohort, together with the observation by Aw and colleagues of cases in which severely impaired posterior SCC gain (<0.20) was found in participants with INO that was only detectable with computational analysis, raises the possibility of the presence of clinically occult INO [18] in at least some of our participants.

Previously, we postulated that the failure to generate CSs during lateral head impulses in the presence of markedly reduced VOR gains may be pathognomonic for INO [18]. Our observation of this same phenomenon in response to vertical head impulses (Figure 1) lends further support to this position as INO is known to affect pursuit and saccadic vertical eye movements [44]. Further research that includes a computerized computational analysis of the eye movements in persons known to have both clinical and brain imaging evidence of INO is needed to confirm or refute this hypothesis. It is plausible that lesions of the medial longitudinal fasciculus may also explain the other atypical oculomotor behaviors that we observed (Figure 2 and Figure 3) as patients with INO have been shown to have deficits in the regulation of vertical gaze holding and vertical gaze-evoked nystagmus [45].

### 4.3. Association between Compensatory Saccades and Disability

Our findings related to the generation of CSs to reduce GPE are more challenging to interpret. The small numbers of PLW-min- and PLW-mod-MS and the low number of CSs evoked with any given head impulse may have affected our power to detect between-group differences. A greater disability was associated with either a higher or lower CS frequency, depending on the SCCs. This is inconsistent with our finding of the lowest VOR gains for the vertical SCCs, as well as a prior study in PLW-MS that showed that an increased CS frequency was associated with lower VOR gains [7]. A predominance of greater right-sided involvement in our cohort (as reflected in the VOR gains), particularly for PLW-mod-MS, may have also skewed our results. Further, the failure of some participants with low VOR gains to generate any CSs with head impulses influenced our calculations of the CS frequency for those canals. Further research in PLW-MS with and without definitive evidence of INO is needed to provide a clearer picture of how compensatory oculomotor behaviors change as MS progresses and as lesions affect specific central nervous system structures.

### 4.4. A Role for Personalized Vestibular Physical Therapy

Vestibular physical therapy (VPT) is an effective intervention for persons with peripheral vestibular hypofunction [46]. Gaze stabilization exercises are a critical component of VPT [47]; however, although recent evidence suggests that VPT is beneficial for PLW-MS, the available literature has not reported on the effect of VPT on gaze stabilization in this population. In a randomized controlled trial of the efficacy of a 14-week programmatic VPT intervention involving 38 PLW-MS who were able to walk 100 m with or without a single-sided device, Herbert and colleagues demonstrated positive effects on fatigue, the upright postural balance, and perceived disability due to dizziness in the intervention group [48]. Subsequently, Ozgen and colleagues demonstrated that 20 PLW-MS with a median EDSS score of 3.5 who were randomized to receive comprehensive VPT for eight weeks showed significant improvements in the measures of the severity of dizziness and imbalance, disability due to dizziness, balance-related confidence, standing balance, and gait [49]. VPT also appears to be effective in severely disabled PLW-MS (EDSS score 6.0 to 7.0) as Tramontano and colleagues found positive effects of a 4-week inpatient intervention on the perception of fatigue, standing balance, short-distance ambulation, and the performance of activities of daily living [50]. Increasing engagement in movement and dynamic functional exercises may also be beneficial, as shown by Loyd and colleagues, who recently found that both a program of focused gaze and postural stability exercises and a program of strength and endurance exercises led to similar reductions in self-reported dizziness in PLW-MS [26]. Based on these prior studies and our findings, PLW-MS, regardless of their level of disability, may be candidates for personalized VPT and exercise prescription that includes interventions to enhance their gaze stabilization with particular emphasis on addressing gaze instability during vertical head movements.

### 4.5. Limitations

This study has several limitations. First, though our cohort was similar, if not larger, in size to the previously reported cohorts, the number of participants is small and the sizes of each group were dissimilar. Second, since we analyzed the vHIT results for PLW-MS who reported dizziness and/or imbalance, our findings may not be generalizable to all PLW-MS. Third, although all participants had evidence of brainstem and/or cerebellar dysfunction in their EDSS Functional Systems Scores, brain imaging was not available to confirm the location and extent of the MS lesions. Fourth, as expected for an often-progressive neurological disease, the duration of the disease differed across the EDSS subgroups; thus, the effects of the learned behavioral oculomotor compensations may differ as a result of a variation in the amount of time spent coping with this chronic health condition. Fifth, as is common in clinical practice, we recorded eye movements with monocular vHIT, thus, the asymmetrical effects of any occult INO on oculomotor behavior cannot be appreciated in our analyses. Larger studies that include a healthy control group and/or a separate group of PLW-MS who do not report dizziness and/or imbalance, use well-balanced group assignment in which the location and extent of the central nervous system’s involvement are confirmed with imaging, employ a binocular recording of the eye movements, and control for the potential differences related to aging and disease duration are needed. Such studies would help further our understanding of the graduated effects of advancing disease on the vestibular function and compensatory oculomotor behavior in PLW-MS.

## 5. Conclusions

Greater disease-related disability is associated with a degradation in the vestibular function and changes in compensatory oculomotor behavior in PLW-MS, however, the EDSS classification schema is insufficient for aiding the characterization of the impact of MS on the gaze stabilization mechanisms. The strategies employed by PLW-MS to recover their gaze stability following rapid head movements appear to be idiosyncratic and dependent upon the specific location and extent of the lesions within the brainstem and cerebellar nuclei and pathways that convey SCC afference. PLW-MS who report dizziness and/or imbalance and who have abnormalities on vHIT may be candidates for VPT. Additionally, the further characterization of compensatory oculomotor behavior in PLW-MS may inform the development of personalized approaches to gaze stabilization interventions in the affected persons.

## Figures and Tables

**Figure 1 brainsci-12-01519-f001:**
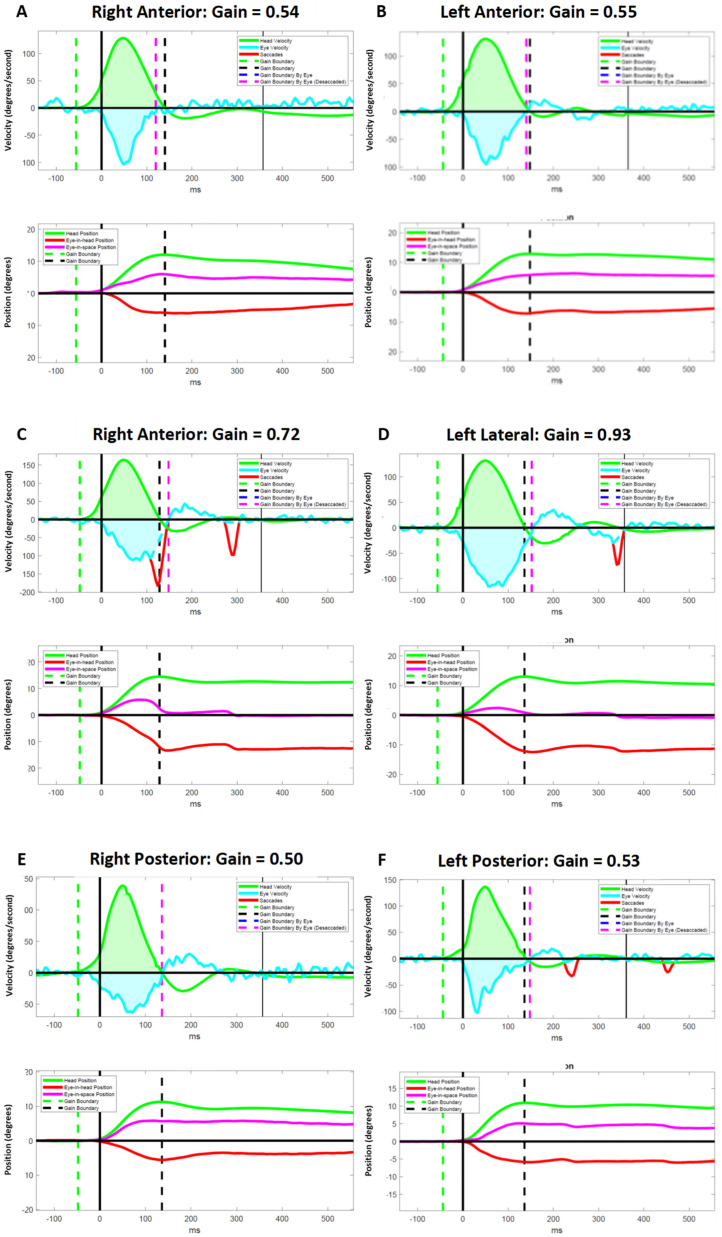
Representative video head impulse test results for a participant with moderate MS who failed to generate effective compensatory saccades in response to vertical head impulses despite having abnormal vestibulo–ocular reflex (VOR) gains for the vertical canals. This subject did generate compensatory saccades with lateral impulses despite having better VOR gains for the right and left lateral canals. Data for the anterior (**A**,**B**), lateral (**C**,**D**), and posterior (**E**,**F**) canals are shown in the velocity (upper panels) and position (lower panels) domains. ms = milliseconds.

**Figure 2 brainsci-12-01519-f002:**
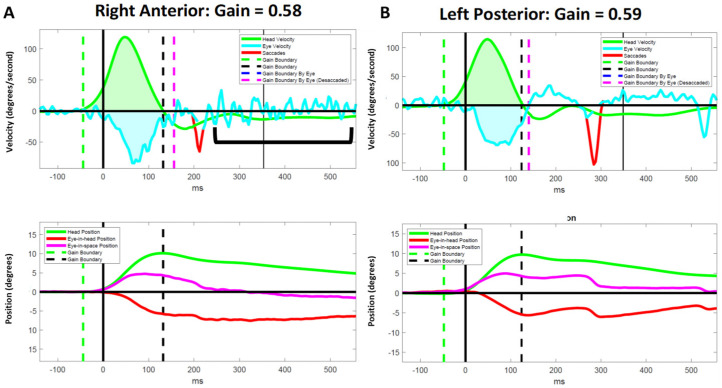
Representative results for a participant with moderate MS who demonstrated an exacerbation of vertical nystagmus [bracket] following right-anterior impulses, but not left-posterior impulses. The presence of nystagmus was confirmed by reviewing the video recording of their vHIT (see Appendix A). Data for the right anterior (**A**) and left posterior (**B**) canals are shown in the velocity (upper panel) and position (lower panel) domains. ms = milliseconds.

**Figure 3 brainsci-12-01519-f003:**
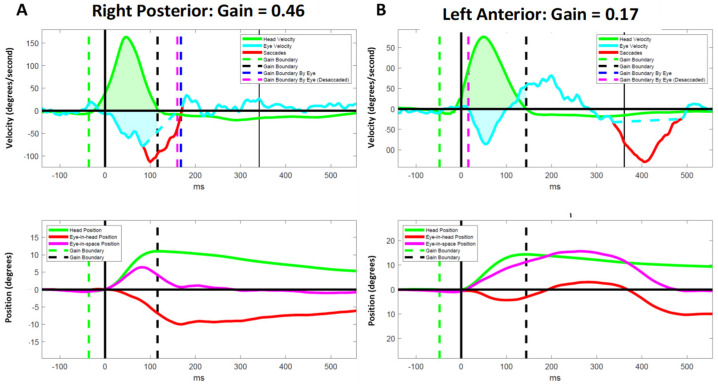
Representative results for a participant with moderate MS who demonstrated large amplitude, low velocity compensatory eye movements with right-posterior and left-anterior impulses in the presence of low vestibulo–ocular reflex gains. Data for the right anterior (**A**) and left posterior (**B**) canals are shown in the velocity (upper panel) and position (lower panel) domains. ms = milliseconds.

**Table 1 brainsci-12-01519-t001:** Participant Characteristics.

Variable	PLW-min-MS (N = 8)	PLW-mild-MS (N = 23)	PLW-mod-MS (N = 6)
Age (years) ^a^	47.3 (12.2)	55.5 (11.9)	54.2 (13.8)
Female sex ^b^	6 (75.0%)	18 (78.3%)	4 (66.7%)
Age at onset (years) ^c^	41.0 (33.8–47.2)	32.0 (28.0–42.5)	38.5 (26.5–46.0)
MS duration (years) ^c^	2.0 (1.0–9.2)	21.0 (11.0–28.5)	18.5 (13.5–19.8)

These data are presented as the mean (standard deviation) ^a^, number (percent) ^b^, or median (inter-quartile range) ^c^. MS = multiple sclerosis. PLW-min-MS = people living with minimal MS. PLW-mild-MS = people living with mild MS. PLW-mod-MS = people living with moderate MS.

**Table 2 brainsci-12-01519-t002:** Between-group Differences in VOR Gain, CS Frequency, CS Latency, and GPE.

Variable	PLW-min-MS (N = 8)	PLW-mild-MS (N = 23)	PLW-mod-MS (N = 6)	*p*-Value
Right lateral canal				
VOR gain	*n* = 66: 0.98 (0.93–1.02)	*n* = 275: 0.91 (0.79–0.99)	*n* = 74: 0.91 (0.79–1.03)	<0.001
CS frequency	*n* = 66: 1.00 (0.00–1.75)	*n* = 275: 1.00 (1.00–2.00)	*n* = 74: 1.00 (1.00–2.00)	0.002
CS latency (ms)	*n* = 42: 230.00 (185.00–283.25)	*n* = 213: 220.00 (168.00–296.00)	*n* = 65: 200.00 (160.00–257.33)	0.152
GPE (°)	*n* = 66: 0.18 (−0.20–1.01)	*n* = 275: 1.16 (0.10–2.72)	*n* = 74: 0.26 (−0.95–1.57)	<0.001
Left lateral canal				
VOR gain	*n* = 79: 0.92 (0.88–0.96)	*n* = 285: 0.86 (0.76–0.93)	*n* = 84: 0.92 (0.84–1.00)	<0.001
CS frequency	*n* = 79: 1.00 (1.00–1.00)	*n* = 285: 1.00 (1.00–2.00)	*n* = 84: 1.00 (1.00–2.00)	0.013
CS latency (ms)	*n* = 63: 244.00 (210.33–308.00)	*n* = 231: 256.00 (213.00–312.33)	*n* = 76: 276.00 (223.50–324.00)	0.085
GPE (°)	*n* = 79: 1.30 (0.52–1.97)	*n* = 285: 2.09 (0.85–3.60)	*n* = 84: 0.84 (−0.12–1.81)	<0.001
Right anterior canal				
VOR gain	*n* = 30: 0.80 (0.65–0.92)	*n* = 151: 0.81 (0.71–1.01)	*n* = 45: 0.66 (0.58–0.77)	<0.001
CS frequency	*n* = 30: 2.00 (1.00–2.00)	*n* = 151: 1.00 (1.00–2.00)	*n* = 45: 0.00 (0.00–1.00)	<0.001
CS latency (ms)	*n* = 27: 140.00 (90.00–206.00)	*n* = 127: 222.00 (140.00–280.00)	*n* = 22: 174.67 (100.00–213.50)	0.001
GPE (°)	*n* = 30: 1.43 (0.85–2.69)	*n* = 151: 1.72 (−0.19–3.44)	*n* = 45: 3.50 (2.33–4.31)	<0.001
Left posterior canal				
VOR gain	*n* = 25: 0.89 (0.75–1.04)	*n* = 161: 0.67 (0.39–0.84)	*n* = 55: 0.52 (0.43–0.73)	<0.001
CS frequency	*n* = 25: 1.00 (1.00–2.00)	*n* = 161: 1.00 (1.00–2.00)	*n* = 55: 1.00 (1.00–2.00)	0.816
CS latency (ms)	*n* = 19: 166.00 (103.00–210.00)	*n* = 138: 211.00 (172.00–275.50)	*n* = 44: 184.33 (92.00–255.50)	0.001
GPE (°)	*n* = 25: 0.79 (−0.36–1.77)	*n* = 161: 4.32 (2.02–6.75)	*n* = 55: 5.02 (2.77–6.31)	<0.001
Left anterior canal				
VOR gain	*n* = 44: 0.85 (0.80–0.93)	*n* = 176: 0.75 (0.62–0.86)	*n* = 51: 0.67 (0.53–0.77)	<0.001
CS frequency	*n* = 44: 1.00 (0.00–1.00)	*n* = 176: 1.00 (0.00–2.00)	*n* = 51: 1.00 (1.00–2.00)	0.002
CS latency (ms)	*n* = 28: 197.00 (115.50–294.00)	*n* = 123: 228.00 (152.00–316.00)	*n* = 42: 159.67 (100.00–214.00)	0.003
GPE (°)	*n* = 44: 1.65 (0.48–2.30)	*n* = 176: 3.00 (1.19–4.78)	*n* = 51: 3.18 (1.92–5.37)	<0.001
Right posterior canal				
VOR gain	*n* = 48: 0.75 (0.71–0.81)	*n* = 191: 0.60 (0.29–0.74)	*n* = 56: 0.56 (0.45–0.70)	<0.001
CS frequency	*n* = 48: 1.00 (1.00–2.00)	*n* = 191: 1.00 (1.00–2.00)	*n* = 56: 1.00 (1.00–2.00)	0.03
CS latency (ms)	*n* = 46: 220.00 (130.00–271.00)	*n* = 160: 209.33 (172.00–308.00)	*n* = 51: 200.00 (128.00–262.00)	0.086
GPE (°)	*n* = 48: 2.84 (1.83–3.61)	*n* = 191: 4.99 (2.50–8.53)	*n* = 56: 3.82 (2.57–5.18)	<0.001

These results are presented as the median (inter-quartile range). *p*-values are from omnibus Kruskal–Wallis tests. Pairwise comparisons are reported in the text. MS = multiple sclerosis. PLW-min-MS = people living with minimal MS. PLW-mild-MS = people living with mild MS. PLW-mod-MS = people living with moderate MS. CS = compensatory saccade. ° = degrees. GPE = gaze position error. N = number of subjects. *n* = number of traces available for the analysis. ms = milliseconds. VOR = vestibulo–ocular reflex.

## Data Availability

The dataset will be made available upon written request to the corresponding author.

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
