# Peer review of "Greater Disability Is Associated with Worse Vestibular and Compensatory Oculomotor Functions in People Living with Multiple Sclerosis"

_brainsci, 2022, doi:10.3390/brainsci12111519_

Round 1

Reviewer 1 Report

The paper “Vestibular and Compensatory Oculomotor Functions Decline 2 as Multiple Sclerosis Progresses “ is aimed to “characterize visual-vestibular function in PLW-MS using six-canal vHIT results and to compare vestibular function and compensatory oculomotor behavior across groups of PLW-MS who have differing levels of disability “. The manuscript is well written, and covers an important unmet need: measurable outcomes that correlate with MS disease worsening and progression.

I have a couple of major concern:

-       The authors divided the population in 3 different disability group and HIT outcomes have been associated with disability. In the title and the text sometimes it seems that HITs outcomes are associated with MS progression. It should be clarified.

-       Did the authors also have tried to associate HIT outcomes with MS disease progressive course? An association with disease MS progression (progressive disease) will be very interesting.

-       The population in study is limited to patients who complained dizziness, impaired balance or 2 falls in the past year: it should be better discussed the external validity of the study, especially in a multifocal CNS disease, with variable involvement of brainstem.

Minor:

-       English language is fine, but some corrections are need: e.g. data is plural and when it is the subject, verbs should be conjugated as plural..

Author Response

Please note that our response to both reviewers are included in a single upload.

Reviewer 2 Report

Grove and colleagues reported on vestibular function and compensatory oculomotor behaviours in multiple sclerosis, also in relation to disability. The topic is interesting, with dizziness being a common issue in multiple sclerosis, and often difficult to address. Methods for data acquisition are sound. However, I would suggest the authors improve their population and statistical design. I have included full suggestions below.

The introduction is too long and very hard to read. I would suggest the authors shorten this as much as possible. For instance, I do not see any need to explain what the EDSS is (can be shortened and included in the methods, if anything). Similarly, I would not include previous study interpretation, but rather findings and limitations. It is indeed difficult to understand the actual novelty of this study.

Based on the EDSS, authors classified patients into minimal, mild or moderate MS. However, this classification is not appropriate, nor validated. There are a number of MS-related issues which are not sufficiently weighted in the EDSS. Thus, I would suggest authors always refer to the EDSS range when naming their subgroups.

The manuscript would have benefited from a control group, since ageing can also affect vestibular function. Also, a group of people with MS without reported dizziness would have been interesting. Based on the present results, you are simply stating that, among people with MS with dizziness, those with higher EDSS also have higher vestibular impairments. Please, notice that vertigo, dizziness and unbalance contribute to EDSS calculation!

There is a huge number of statistical tests and results. I would suggest authors use conservative statistical approaches to reduce the number of tests and highlight more significant findings.

Compensatory saccades are conventionally thought to indicate peripheral vestibular dysfunction (e.g., HINTS). I was wondering how authors would explain their presence in a disease of the central nervous system.

In the main text, VOR is not spelled out on its first appearance.

Author Response

Please note that our responses to both reviews are on the same word .doc

Round 2

Reviewer 1 Report

The authors clarified all my concerns. I suggest to accept the paper in the present form

Reviewer 2 Report

Authors have addressed my concerns.